# The Rhodanese PspE Converts Thiosulfate to Cellular Sulfane Sulfur in *Escherichia coli*

**DOI:** 10.3390/antiox12051127

**Published:** 2023-05-20

**Authors:** Qiaoli Yu, Mingxue Ran, Yuping Xin, Huaiwei Liu, Honglei Liu, Yongzhen Xia, Luying Xun

**Affiliations:** 1State Key Laboratory of Microbial Technology, Shandong University, 72 Binhai Road, Qingdao 266237, China; 2School of Molecular Biosciences, Washington State University, Pullman, WA 991647520, USA

**Keywords:** sulfane sulfur, thiosulfate, rhodanese, *Escherichia coli*

## Abstract

Hydrogen sulfide (H_2_S) and its oxidation product zero-valent sulfur (S^0^) play important roles in animals, plants, and bacteria. Inside cells, S^0^ exists in various forms, including polysulfide and persulfide, which are collectively referred to as sulfane sulfur. Due to the known health benefits, the donors of H_2_S and sulfane sulfur have been developed and tested. Among them, thiosulfate is a known H_2_S and sulfane sulfur donor. We have previously reported that thiosulfate is an effective sulfane sulfur donor in *Escherichia coli*; however, it is unclear how it converts thiosulfate to cellular sulfane sulfur. In this study, we showed that one of the various rhodaneses, PspE, in *E. coli* was responsible for the conversion. After the thiosulfate addition, the ΔpspE mutant did not increase cellular sulfane sulfur, but the wild type and the complemented strain ΔpspE::pspE increased cellular sulfane sulfur from about 92 μM to 220 μM and 355 μM, respectively. LC-MS analysis revealed a significant increase in glutathione persulfide (GSSH) in the wild type and the ΔpspE::pspE strain. The kinetic analysis supported that PspE was the most effective rhodanese in *E. coli* in converting thiosulfate to glutathione persulfide. The increased cellular sulfane sulfur alleviated the toxicity of hydrogen peroxide during *E. coli* growth. Although cellular thiols might reduce the increased cellular sulfane sulfur to H_2_S, increased H_2_S was not detected in the wild type. The finding that rhodanese is required to convert thiosulfate to cellular sulfane sulfur in *E. coli* may guide the use of thiosulfate as the donor of H_2_S and sulfane sulfur in human and animal tests.

## 1. Introduction

Sulfur is essential for all organisms [1]. Its presence and functions in amino acids, cofactors, and polysaccharides are well known. New functions of sulfur in biology have been recognized in the form of hydrogen sulfide (H_2_S) and sulfane sulfur. H_2_S is considered a gasotransmitter, similar to carbon monoxide [2] and nitric oxide [3,4]. It exerts various physiological benefits, such as cytoprotection, anti-inflammation, angiogenesis, and vasodilation [3]. H_2_S is oxidized to zero-valent sulfur (S^0^). Inside cells, S^0^ often exists in different forms, including hydrogen polysulfide (H_2_S_n_, n ≥ 2), organic polysulfide (RSS_n_H, RSS_n_R, n ≥ 2), and octasulfur (S_8_) [5,6], which are collectively referred to as sulfane sulfur [7]. Thiosulfate anion (S_2_O_3_^2−^) is a stable form of sulfane sulfur, but glutathione persulfide (GSSH) and S_8_ are reactive [8]. The cellular activities of H_2_S are often mediated through sulfane sulfur, including signaling, redox homeostasis, and metabolic regulations [9].

Sulfane sulfur and H_2_S often coexist inside cells. Sulfane sulfur is produced from either H_2_S oxidation or L-cysteine metabolism [10,11,12,13]. Humans and some heterotrophic bacteria possess a H_2_S oxidation pathway [14,15,16,17,18]. Sulfide: quinone oxidoreductase (SQR) oxidizes H_2_S to sulfane sulfur, such as GSSH, and persulfide dioxygenase (PDO) oxidizes GSSH to sulfite [17,19,20,21,22]. Sulfite spontaneously reacts with sulfane sulfur to produce thiosulfate, or the reaction is catalyzed by rhodanese (RHOD) [23,24]. Since PDOs have high *K_m_* values for GSSH [25], their role is likely to prevent the excessive accumulation of cellular sulfane sulfur. Besides H_2_S oxidation, sulfane sulfur can be produced from cysteine by various enzymes, including cysteinyl-tRNA synthetase, 3-mercaptopyruvate sulfurtransferase, and cystathionine γ-lyase [10,11,12,13]. In cells without PDO, cellular sulfane sulfur is also reduced by cellular thiols to H_2_S: for example, GSSH reacts with glutathione (GSH) to form GSSG and H_2_S [26,27]. The continuous production and metabolism of sulfane sulfur and H_2_S may keep them in a certain range.

Various H_2_S donors have been developed and tested in animal models as therapeutic agents [28,29,30]. Among them, thiosulfate has been tested in clinical trials [31], because it is a clinically approved drug to treat cyanide poisoning [32], calciphylaxis—a disease involving deposits of calcium salts in blood vessels in the dermis and subcutaneous tissue [33]—and cisplatin overdose [34]. Thiosulfate is the main end product of sulfide oxidation in human mitochondria and various heterotrophic bacteria under aerobic conditions [16,35]. Under hypoxic conditions, thiosulfate is reduced to H_2_S by RHOD and 3-mercaptopyruvate sulfurtransferase (3MST) at the expense of dihydrolipoamide [36,37]. For bacteria with the cysteine synthase-b (CysM), they directly use thiosulfate for L-cysteine synthesis [38], and L-cysteine can be metabolized to H_2_S [39]. Microorganisms without CysM deploy an alternative pathway. Thiosulfate is converted by RHOD to GSSH, which is further reduced by another GSH to produce H_2_S, and H_2_S is used by the cysteine synthase-a (CysK) to produce L-cysteine [40]. Thus, thiosulfate has a good potential to be used as a donor of H_2_S and sulfane sulfur in mammals and bacteria.

RHODs were originally referred to as thiosulfate: cyanide sulfurtransferases (E.C.2.8.1.1), as they detoxify cyanide, but now, they are broadly considered to transfer sulfur atoms from compounds containing sulfane sulfur to various nucleophilic acceptors [41,42]. RHODs come in different forms. Some are single-domain proteins, such as *Escherichia coli* PspE [43], YgaP [44], and human TSTD1 [45], and others have two or more RHOD domains, such as bovine RHOD Rhobov [46], *Azotobacter vinelandii* RHOD RhdA [47], and *E. coli* YnjE [44]. *E. coli* contains nine genes (*sseA*, *ynjE*, *glpE*, *ygaP*, *pspE*, *ybbB*, *yibN*, *yceA*, *thiI*) encoding proteins with an RHOD domain that possess the active site consensus sequence Cys-Xaa-Xaa-Gly [48]. When these genes are deleted in *E. coli*, except *thiI*, the mutant RHOD-8K grows similarly to the wild type, but has no detectable RHOD activities [23,24].

Among several tested sulfane sulfur donors, thiosulfate increases cellular sulfane sulfur the most in *E. coli* [49]. It is unclear whether (1) RHOD is responsible for converting thiosulfate to reactive sulfane sulfur, (2) thiosulfate is first used by CysM for cysteine synthesis and then cysteine metabolism leads to increased cellular sulfane sulfur, or (3) another enzyme is responsible for the conversion. Here, we used genetic and biochemical analyses and showed that the RHOD PspE is mainly responsible for rapidly converting stable thiosulfate to cellular sulfane sulfur in *E. coli*.

## 2. Materials and Methods

### 2.1. Bacterial Strains, Culture Conditions, and Reagents

The strains and plasmids used in this work are listed in Appendix A. *Escherichia coli* MG1655 and its mutants were grown in lysogeny broth (LB) at 37 °C. Kanamycin (Km^r^) (50 μg/mL) was added when required. Hydrogen peroxide (H_2_O_2_) was purchased from Bio Basic Inc (Markham, ON, Canada). Thiosulfate, GSH, and monobromobimane (mBBr) were purchased from Sigma-Aldrich (Burlington, MA, USA). Sulfane sulfur probe 4 (SSP4) was purchased from Dojindo Molecular Technologies, Inc. (Tokyo, Japan). N-Iodoacetyl-L-tyrosine methyl ester (TME-IAM) was purchased from HEOWNS (Tianjin, China). Lead (II) acetate trihydrate and other chemicals were purchased from Sangon Biotech (Shanghai, China).

### 2.2. Construction of RHOD Mutants

Gene deletion mutants of *E. coli* MG1655 were generated according to the reported Datsenko and Wanner method [50] with long homology arms to increase efficiency [51]. *E. coli* MG1655 with the pTKred plasmid was induced with isopropyl β-D-1-thiogalactopyranoside (IPTG) to express the red recombinase genes. The induced cells were harvested and prepared as competent cells. PCR products consisting of the kanamycin resistance gene (Km^r^) and FRT sites flanked by the ends of single RHOD-encoding genes were electroporated into the competent cells and cultured in LB medium at 37 °C for 2 h. The cells were then spread onto Km^r^ LB plates and incubated at 37 °C overnight. The pCP20 plasmid was transferred into the correct double-exchange cells to remove the Km^r^ sequence via recombination at the FRT sites. The plasmids and primers used are listed in Appendix A. Five genes encoding reported RHOD activities were deleted as the single-deletion mutants Δ*pspE*, Δ*glpE*, Δ*ygaP*, Δ*ynjE*, and Δ*sseA*, or as the five-deletion mutant RHOD-5K (*sseA*, *pspE*, *glpE*, *ynjE*, *ygaP*). The order of gene deletion was s*seA*, *glpE*, *pspE*, *ynjE*, and *ygaP* to generate RHOD-5K. The eight-gene-deletion mutant RHOD-8K (*sseA*, *pspE*, *glpE*, *ynjE*, *ygaP*, *yceA*, *yibN*, *ybbB*) has been previously reported [24]. The complemented strains carry genes cloned in pBBR1MCS2 [52].

### 2.3. Growth Curves

Fresh colonies of *E. coli* were inoculated in 4 mL of LB medium and grown with shaking at 200 rpm at 37 °C overnight. The cells were transferred to 400 µL of fresh LB medium at an initial OD_600nm_ of 0.05 in a 48-well plate. Thiosulfate and H_2_O_2_ were added as indicated in the text. The cultures were incubated with shaking at 37 °C, and the OD_600nm_ was read every 30 min in a plate reader (BioTek Synergy H1, Agilent, Santa Clara, CA, USA).

### 2.4. Protein Expression and Purification

The genes encoding PspE and YnjE without the corresponding N-terminal signal sequence, ΔNS-*pspE* (258 bp) and ΔNS-*ynjE* (1239 bp), and the intact genes of *glpE* (324 bp) and *yceA* (1050 bp) were separately amplified from the *E. coli* MG1655 genome with the primers listed in Table 1. Each gene was assembled into pET30a by using the T5 exonuclease-dependent assembly method [53]. The recombinant proteins contained a C-terminal His×6-tag derived from the vector. The *E. coli* BL21(DE3) strain harboring the expression plasmid was incubated in LB medium with shaking (200 rpm) at 37 °C. When the OD_600nm_ reached 0.8, 0.4 mM isopropyl β-D-1-thiogalactopyranoside (IPTG) was added to induce expression, and the culture was further incubated with shaking (150 rpm) at 20 °C for 20 h. Then, cells were harvested via centrifugation at 4 °C and resuspended in buffer I (20 mM Tris-HCl, 0.5 M NaCl, 20 mM imidazole, pH 8.0). Cell disruption was performed using a Pressure Cell Homogeniser (SPCH-18, Harlow, Essex, UK) at 4 °C. The cell lysate was centrifuged to remove the debris. The target protein in the supernatant was purified by passing it through a nickel nitrilotriacetate (Ni-NTA) agarose column and a size exclusion column (Superdex 200; GE Healthcare, Shanghai, China). The purified protein was stored in 50 mM Tris-HCl (pH 7.4) containing 6% glycerol at −80 °C.

### 2.5. Thiosulfate: GSH Sulfurtransferase Activity Assay

When RHOD uses thiosulfate and GSH as the substrates, it is referred to as thiosulfate: GSH sulfurtransferase, producing GSSH, and sulfite GSSH spontaneously reacts with another GSH to generate GSSG and H_2_S. H_2_S was detected by including lead acetate in the reaction mixture, as previously described [54]. Briefly, the reaction mixture (1 mL) contained 0.4 mM lead acetate, 0.5 mM~150 mM thiosulfate, and 1 mM~150 mM GSH in 100 mM HEPES buffer (pH 7.4). A purified RHOD was added to initiate the reaction. The RHODs were added in different amounts, depending on the enzyme activity. An amount of 5 μg of ΔNS-PspE, 5 μg of GlpE, 100 μg of YceA, or 200 μg of ΔNS-YnjE was used in the 1 mL reactions. The reaction was incubated at 37 °C for 8 min. Lead sulfide was measured at 390 nm, and its concentration was calculated with an extinction coefficient of 5500 M^−1^ cm^−1^ [54].

### 2.6. HPLC Analysis of Total Cellular Sulfane Sulfur

Colonies of *E. coli* were collected and cultured in LB medium overnight. The cultures were transferred into fresh LB medium at an initial OD_600nm_ of 0.05. Cells were cultured at 37 °C until the OD_600nm_ reached 1.0, harvested via centrifugation, and resuspended in phosphate-buffered saline (PBS) buffer to an OD_600nm_ of 2.0. Thiosulfate was added to the cell culture at room temperature for 1 h. Total cellular sulfane sulfur was determined according to a reported method [49]. Briefly, the cells were harvested via centrifugation, washed once with 50 mM Tris buffer (pH 7.4), and resuspended in 50 mM Tris buffer (pH 9.5) with 1 mM sulfite or 0.5 mM DTT. The sample was heated at 95 °C for 10 min, during which cellular sulfane sulfur reacted with sulfite to produce thiosulfate. The produced thiosulfate was derivatized with mBBr and analyzed using HPLC.

### 2.7. SSP4 Staining for Sulfane Sulfur Detection

SSP4 is a fluorescent probe used to specifically detect sulfane sulfurs [8]. Resting *E. coli* cells in PBS at an OD_600nm_ of 2.0 were incubated with 20 μM SSP4 at 37 °C in the dark for 1 or 2.3 h, and the produced fluorescence was detected with an excitation of 482 nm and emission of 515 nm using a Synergy H1 microplate reader (Synergy H1, BioTek, Winooski, VT, USA). CTAB was added at 25 or 500 μM, of which 25 μM was used to speed up the SSP4 reaction and 500 μM was used to permeabilize the cell membrane. Thiosulfate was added at 10 mM as indicated in the text.

### 2.8. GSSH Preparation and Detection

For the preparation and detection of GSSH, 17 mM GSH in 50 mM Tris-HCl (pH 8.0) and 50 μM DTPA was mixed with 17 mM S_8_ in acetone in equal volumes. Immediately, the mixture was centrifuged at 13,000× *g* for 1 min. The supernatant was saved, and GSSH was determined using a cyanide method [18]. Briefly, 250 μL of GSSH solution was added to a mixture of 550 μL of 1% boric acid and 200 μL of 100 mM cyanide and heated in boiling water for 2 min. After cooling to room temperature, 100 μL of ferric nitrate reagent was added. The sample was centrifuged at 13,000× *g* for 3 min, and the supernatant was detected at 460 nm for absorbance.

### 2.9. Lead Acetate Strips for the Detection of Hydrogen Sulfide

Colonies were cultured overnight in LB medium, and the cultures were transferred to fresh LB medium at an initial OD_600nm_ of 0.05 and cultured to an OD_600nm_ of 1.0. Then, 10 mM thiosulfate was added, and a lead acetate test strip was placed in the gas phase. The cultures were incubated at 37 °C with shaking at 200 rpm for 1, 6, and 20 h. The darkness of the lead acetate on the strip was observed.

### 2.10. HPLC Analysis for Sulfide Detection

Colonies were cultured overnight in LB medium, and the cultures were transferred to fresh LB medium at an initial OD_600nm_ of 0.05 and cultured to an OD_600nm_ of 1.0. Then, 10 mM thiosulfate was added. The cultures were incubated at 37 °C with shaking at 200 rpm for 1 h. Afterwards, 1 mL of the culture was centrifuged at 13,000× *g* for 3 min at 4 °C. Supernatants were taken to react with mBBr for half an hour, and the sulfide-bimane adduct was detected using HPLC [5]. The standard curve of sulfide was generated in fresh LB medium.

### 2.11. LC-MS/MS Analysis of PspE-Cys49

The ΔNS-PspE protein (1.35 mg/mL, 5 μg) was reacted with 10 mM thiosulfate at room temperature for 30 min. Denaturing buffer (30 mM Tris-HCl, 8 M urea, pH 8.0) and excess N-Iodoacetyl-L-tyrosine methyl ester (TME-IAM) were added to denaturalize the protein and block free thiols. The protein was then digested with trypsin (Promega) at 37 °C for 20 h, and the peptides passed through a C18 Zip-Tip (Millipore) for desalting before being analyzed via HPLC-tandem mass spectrometry (LC-MS) using a Prominence nano-LC system (Shimadzu, Kyoto, Japan) and LTQ-OrbitrapVelos Pro CID mass spectrometer (Thermo Scientific, Walthan, MA, USA). Full-scan MS spectra (from 300 to 1500 *m*/*z*) were detected at a resolution of 60000 at 400 *m*/*z*.

### 2.12. LC-MS Analysis of Cellular Sulfane Sulfur Species

*E. coli* strains in LB medium were cultured at 37 °C until an OD_600nm_ of 1.0 and harvested via centrifugation. The cells were resuspended in PBS with 10 mM thiosulfate at an OD_600nm_ of 2.0, and incubated at 37 °C for 1 h. The cells were harvested via centrifugation and resuspended with 40 mM Tris-HCL solution (pH 7.4) in 80% methanol at an OD_600nm_ of 6.5. After adding 5 mM N-iodoacetyl-L-tyrosine methyl ester (TME-IAM), the mixture was frozen in liquid nitrogen and thawed in a water bath at 37 °C. The freeze–thaw cycle was repeated six times. The sample was further incubated at 37 °C in the dark for 1 h to complete the derivatization. An equal volume of 0.1% formic was added to the derivatization solution. The sample was centrifuged at 16,200× *g* for 10 min. An amount of 5 μL of the supernatant was injected into a reverse-phase C18 column (VP-ODS, 150 × 4 mm, Shimadzu, Kyoto, Japan). The following gradient of solvent A (0.25% acetic acid and 10% methanol) and solvent B (0.25% acetic acid and 90% methanol) was used: 8% B to 40% B in 7 min, 40% B for 5 min, 40% B to 100% B in 0.1 min, and 100% B for 6 min at a flow rate of 0.8 mL/min. A mass spectral detector (AB SCIEX, Framingham, MA, USA) was used to detect the derivatized persulfide and polysulfide.

## 3. Results

### 3.1. PspE was the Main Enzyme Generating Sulfane Sulfur from Thiosulfate

Among the eight genes encoding potential RHODs, five code for proteins with reported RHOD activities (*sseA*, *ynjE*, *glpE*, *ygaP*, and *pspE*). The five genes were individually deleted to create five single-deletion mutants (ΔsseA, ΔynjE, ΔglpE, ΔygaP, ΔpspE), and they were all deleted in a single mutant, RHOD-5K. The *E. coli* mutant RHOD-8K, with all eight genes deleted, has no detectable RHOD activities [43]. The *cysM* deletion mutant (ΔcysM) has been previously reported [39]. The wild type and its mutants had similar levels of sulfane sulfur in the absence of thiosulfate (Figure 1). When thiosulfate was added, the wild type and most mutants, except for RHOD-8K, RHOD-5K, and ΔpspE, produced increased cellular sulfane sulfur (Figure 1). PspE is a periplasmic RHOD [55]. The intact PspE or PspE without the N-terminal signal peptide (ΔNS-PspE) that was in the cytoplasm complemented ΔpspE in converting thiosulfate to sulfane sulfur (Figure 1). The cellular sulfane sulfur (nmol. mL^−1^. OD^−1^) in Figure 1 was used to estimate the cellular concentrations by using a reported conversion factor: 1 mL of *E. coli* suspension at an OD_600nm_ of 1 has a 3.6 μL cell volume [56]. Without the added thiosulfate, the *E. coli* wild type and its mutants had similar cellular sulfane sulfur concentrations, with 92 μM on average. The wild type, ΔpspE, and ΔpspE::pspE with thiosulfate contained 220, 89, and 355 μM cellular sulfane sulfur.

Whether other RHODs converted thiosulfate to cellular sulfane sulfur was tested in the *E. coli* RHOD-8K mutant by using a time course experiment. The wild type converted thiosulfate to cellular sulfane sulfur, but RHOD-8K did not (Figure 2A,B). Besides RHOD-8K::pspE (Figure 2C), RHOD-8K::glpE also produced increased cellular sulfane sulfur in the presence of thiosulfate (Figure 2D). When thiosulfate was added, RHOD-8K::sseA and RHOD-8K::ynjE showed increased cellular sulfane sulfur, but the increase was small with overlapping error bars (Figure 2E,F); RHOD-8K::ygaP, RHOD-8K::yceA, RHOD-8K::ybbB, and RHOD-8K::yibN did not increase cellular sulfane sulfur (Figure 2G–J).

Increased cellular sulfane sulfur might lead to its reduction by cellular thiols to H_2_S, which rapidly occurs when elemental sulfur is added to *E. coli* suspensions [57]. Under the same conditions as in Figure 1, increased H_2_S in the gas phase was not apparent in the presence of thiosulfate after incubation for 1 h (Appendix A). A slight increase in H_2_S was only observed in the complemented strains ΔpspE::ΔNS-pspE and ΔpspE::pspE after 20 h of incubation (Appendix A), reflecting their higher cellular sulfane sulfur concentration than that of the wild type (Figure 1). Sulfide in the culture supernatants after incubating with thiosulfate for 1 h was also detected, and no apparent accumulation of sulfide was detected in the wild type, ΔpspE, and its complemented strains ΔpspE::ΔNS-pspE and ΔpspE::pspE (Appendix A). The results suggest that the increased cellular sulfane sulfur from thiosulfate does not lead to a rapid reduction of sulfane sulfur by cellular thiols in *E. coli*.

### 3.2. The Kinetics and Molecular Mechanism of PspE in Its Thiosulfate: Glutathione Sulfurtransferase Activity

We purified several *E. coli* RHODs: ΔNS-PspE, GlpE, ΔNS-YnjE, and YceA with a C-terminal His-tag (Appendix A). Like PspE, YnjE is also a periplasmic protein, and its gene without the signal peptide (ΔNS-ynjE) can be expressed in *E. coli* as a cytoplasmic protein [44,58]. The kinetic parameters of these proteins catalyzing thiosulfate: glutathione sulfurtransferase activities were determined (Appendix A). Both ΔNS-PspE and GlpE demonstrated the activity of catalyzing the reaction between GSH and thiosulfate, but PspE was more effective in catalyzing the reaction than GlpE based on the kinetics (Table 1). The main difference was the *K_m_* values for thiosulfate, as the *K_m_* of ΔNS-PspE for thiosulfate was 7.8 mM and that of GlpE was 33.6 mM. The detailed kinetic parameters are presented in Table 1. ΔNS-YnjE and YceA showed no apparent activity of catalyzing the reaction between thiosulfate and GSH (Table 1). These results show that PspE transfers zero-valent sulfur (S^0^) from thiosulfate to GSH, producing GSSH.

The ΔNS-PspE protein was mixed with thiosulfate and analyzed via LC-MS/MS. A peptide containing the catalytical Cys49 (Cys49-SH) was detected with modifications in the form of Cys49-SSH (Appendix A). In the control sample without thiosulfate, only Cys49-SH was detected (Table 2). These results support that PspE uses the standard “ping-pong” mechanism to capture zero-valent sulfur (S^0^) from thiosulfate and transfer it to GSH. Since only about 1% of the Cys49 thiol was in the persulfide form, Cys49-SSH was unstable under the testing conditions.

### 3.3. The Derived Sulfane Sulfur from Thiosulfate was Mainly Transported into the Cytoplasm

Considering that PspE is a periplasmic protein [55], the resulting sulfane sulfur should originate in the periplasmic space. To test whether sulfane sulfur was exported into the medium, SSP4 was selected [58], as it has poor cell membrane permeability [8]. CTAB is often applied to disrupt the membrane, but it also accelerates the reaction [8]. It was found that 25 μM and 500 μM CTAB enhanced fluorescence production at a similar level (Appendix A); however, 25 μM CTAB did not disrupt the *E. coli* cell membrane, but 500 μM CTAB did, allowing SSP4 to enter the cytoplasm (Appendix A). Subsequently, 20 μM SSP4 and 25 μM CTAB were used to detect extracellular sulfane sulfur, and 20 μM SSP4 and 500 μM CTAB were used to detect intracellular sulfane sulfur. Wild-type MG1655, ΔpspE, ΔpspE::pspE, and ΔpspE::ΔNS-pspE were tested. Extracellular sulfane sulfur was not increased with thiosulfate in the cell suspensions of ΔpspE, wild-type MG1655, and ΔpspE::ΔNS-pspE (Figure 3A–C), but it slightly increased in the suspension of ΔpspE::pspE (Figure 3D). Intracellular sulfane sulfur was increased with thiosulfate in the suspensions of wild-type MG1655, ΔpspE::pspE, and ΔpspE::ΔNS-pspE (Figure 3B–D), but not in the suspension of ΔpspE (Figure 3A).

To verify whether sulfane sulfur was transported into the cells, the cells were harvested and disrupted, and cellular sulfane sulfur species were alkylated with TME-IAM and determined using LC-MS/MS. Large amounts of GSSH (Figure 4A) and several other sulfane sulfur species, including GSSSH, GSSSG, Cys-SSH, H_2_S_2,_ and H_2_S_3_ (Figure 4B–F), were increased in the *pspE* overexpression strain (ΔpspE::pspE) in the presence of thiosulfate. These sulfane sulfur species were also increased in the ΔNS-pspE overexpression strain (ΔpspE::ΔNS-pspE), but the increase was less than that in the *pspE* overexpression strain (ΔpspE::pspE) (Figure 4).

### 3.4. PspE was Responsible for Resisting H_2_O_2_ in the Presence of Thiosulfate in E. coli

H_2_O_2_ inhibits the growth of *E. coli* cells [59]. We tested the growth of the *E. coli* MG1655 wild type, ΔpspE, and ΔpspE::pspE in LB medium. It was found that 10 mM H_2_O_2_ completely inhibited the growth of the *E. coli* strains (Figure 5). When 2 mM thiosulfate was added, the wild type and ΔpspE::pspE had a 6 h lag time and 7 h lag time, respectively, but the *pspE* mutant did not grow (Appendix A). With 10 mM thiosulfate, the MG1655 wild type and ΔpspE::pspE grew after a 2 h lag time, but the *pspE* deletion strain grew after a 4.5 h lag time and grew at a slower rate than the other two strains (Figure 5).

## 4. Discussion

We have previously reported that exogenously introduced thiosulfate significantly increased cellular sulfane sulfur in *E. coli* [49]. There are two possible ways to convert thiosulfate to cellular sulfane sulfur. First, the cysteine synthase CysM may use thiosulfate to produce cysteine [38], and cysteine is then used by cysteine aminotransferase and SseA to produce sulfane sulfur [60]. However, the deletion of *cysM* or *sseA* did not affect the production of cellular sulfane sulfur in the mutants ΔcysM and ΔsseA (Figure 1). These results nullify this possibility. The second way involves the conversion of thiosulfate to cellular sulfane sulfur by RHODs [40]. Our genetic and biochemical analyses unequivocally point to RHODs being responsible for increasing cellular sulfane sulfur from the added thiosulfate.

*E. coli* has nine genes encoding a potential active RHOD domain [43]. Only when *pspE* was deleted did the addition of thiosulfate not increase cellular sulfane sulfur in the mutant ΔpspE (Figure 1). These results raise the question of why other RHODs in *E. coli* are not functional for activity. Since PspE is a periplasmic protein [55], we tested whether the location was responsible. We cloned and expressed the mature protein without the signal peptide (ΔNS-PspE), and ΔNS-PspE was produced as a cytoplasmic protein. The complemented strains ΔpspE::pspE and ΔpspE::ΔNS-pspE both converted thiosulfate to cellular sulfane sulfur (Figure 1), indicating that PspE in either the periplasmic space or the cytoplasm transforms thiosulfate to cellular sulfane sulfur.

The role of PspE in converting thiosulfate to cellular sulfane sulfur could be due to its dominant activity in *E. coli*. GlpE was able to complement ΔpspE in RHOD-8K::glpE when it was supplied on a plasmid and overexpressed (Figure 2). This observation is consistent with its contribution to only a small percentage of RHOD activity in the *E. coli* wild type, in which GlpE represents 85% of the activity and PspE essentially contributes the remainder [43]. Further, PspE is more efficient than GlpE in catalysis, as judged by the catalytic efficiency (k_cat_/K_m_) (Table 1). Other *E. coli* RHODs failed to complement ΔpspE in RHOD-8K (Figure 2), possibly due to a lack of activity (Table 1). Thus, *E. coli* primarily uses PspE to convert thiosulfate to cellular sulfane sulfur

Our results are in agreement with previously reported results. SseA prefers to use 3-mercaptopyruvate as the sulfur donor, and it poorly uses thiosulfate as a substrate [61]. YnjE has poor activity with either 3-mercaptopyruvate or thiosulfate as the sulfur donor, and its physiological substrate is unclear [44]. YceA catalyzes oxygen-dependent 5-deoxyuridine formation in tRNA. YbbB is a tRNA 2-selenouridine synthase that catalyzes the selenophosphate-dependent substitution of selenium for sulfur in 2-thiouridine residues in tRNA [62]. YgaP has a single RHOD domain in the cytoplasm with a C-terminal anchor to the cytoplasmic membrane. Although its RHOD activity was not tested, it can obtain the sulfane sulfur atom from thiosulfate to form a persulfide of its catalytic Cys thiol [63]. The activity of YibN has not been reported. Collectively, most *E. coli* RHODs cannot convert thiosulfate to cellular sulfane sulfur, but PspE and GlpE can.

GSH is the dominant cellular thiol in *E. coli* [64], and PspE may use it as the sulfur acceptor to generate GSSH. When we checked the species of cellular sulfane sulfur, GSSH was the dominant species (Figure 4). GSSH may be transformed to produce other species, such as GSSSH, Cys-SSH, and H_2_S_2_. H_2_S_2_ was the second most abundant species, but its concentration was not increased after PspE converted thiosulfate to cellular sulfane sulfur (Figure 4E). GSH is also common in the periplasmic space, as an ABC-type transporter actively pumps it from the cytoplasm into the periplasmic space [65]. PspE converted thiosulfate to sulfane sulfur in the periplasmic space, and it was then transported into the cytoplasm (Figure 3). The transport mechanism is currently unknown; however, zero-valent sulfur easily enters microbial cells via an unknown mechanism [57]. ΔNS-PspE and GlpE produced sulfane sulfur directly inside the cytoplasm (Figure 2), where GSH is sufficient.

Since the thiosulfate: GSH sulfurtransferase activity of RHODs is strongly inhibited by the product GSSH [45], the reaction normally favors the reverse direction. The functional role of the human RHOD during H_2_S oxidation is to catalyze the reverse reaction for the production of thiosulfate, as the kcat/*K_m_* value for the reverse reaction is 217,000-fold faster than that of the forward reaction [54,66]. The forward reaction can be enhanced by removing GSSH. Upon coupling with a persulfide dioxygenase that oxidizes GSSH to sulfite, the activity is accelerated, and the coupling also lowers RHOD’s *K_m_* value for GSH by ≥25-fold [45]. Despite the thiosulfate: GSH sulfurtransferase activity being unfavorable, it plays an essential role when yeast uses thiosulfate to produce L-Cysteine. The yeast RHOD converts thiosulfate and GSH to GSSH, which is believed to be reduced by GSH to produce H_2_S for L-Cysteine synthesis [40]. Recently, the yeast cysteine synthase has been shown to directly use GSSH for cysteine production [21]. Our results support that PspE converts thiosulfate to cellular sulfane sulfur with GSSH as the dominant species, which then inhibits its further production (Figure 1 and Figure 4). The limited production of GSSH may be responsible for the nonapparent release of H_2_S when thiosulfate was added to the culture medium (Appendix A).

H_2_S has numerous physiological effects. It is a well-known vasorelaxant [66]. In humans with hypertension, H_2_S-generating enzymes including cystathionine γ-lyase are markedly decreased [67]. Inactivation of the H_2_S-producing cystathionine γ-lyase in mice reduced endogenous H_2_S levels, and the mutant mice developed hypertension with an onset at the age of 8 weeks [68]. H_2_S may reduce blood pressure via several mechanisms [69], but the direct effect is the opening of a K^+^ channel by forming a persulfide of a Cys thiol [70], suggesting that H_2_S is oxidized to sulfane sulfur, which modifies the Cys thiol to the persulfide [71,72]. In bacteria, the effects of H_2_S are also usually exerted through sulfane sulfur. H_2_S increases sulfane sulfur in the form of CoA-SSH in *Enterococcus faecalis*, which inhibits phosphotransacetylase that catalyzes the conversion of acetyl-CoA and acetyl-phosphate, a key step of acetate production and utilization [73]. Bacteria with H_2_S-oxidizing genes often use a gene regulator that responds to sulfane sulfur to induce the expression of the H_2_S-oxidizing genes [74]. The growth-associated variations in cellular sulfane sulfur regulate the expression of antibiotic resistance genes controlled by the MarR family regulators [75,76].

H_2_S donors have been tested as therapeutic agents in many cases [77,78]. Some H_2_S donors are sulfane sulfur species. SG-1002 consists of 92% S_8_, which is shown to protect against pressure-overload-induced heart failure [79]. Thiosulfate has been used as a H_2_S donor and shown to attenuate angiotensin II-induced hypertensive heart disease in rats [80]. In *E. coli*, thiosulfate is converted to reactive sulfane sulfur species, including GSSH, by PspE (Figure 4). The increased reactive sulfane sulfur species are responsible for the improved resistance to H_2_O_2_ (Figure 5). Inside the cytoplasm, sulfane sulfur can be reduced by cellular thiols and redox enzymes to H_2_S [26,27]. In humans, the mitochondrial SQR oxidizes H_2_S to sulfane sulfur [22]. Due to the interchanging natures of H_2_S and sulfane sulfur inside cells, both may have similar effects [81]. However, accumulating evidence supports that most physiological effects of H_2_S are exerted via sulfane sulfur through Cys residue persulfidation [72,82]. Our finding that the RHOD PspE converts thiosulfate to increase cellular sulfane sulfur in *E. coli* may guide the use of thiosulfate as a donor of reactive sulfane sulfur in clinical applications.

## 5. Conclusions

Our genetic and biochemical analyses revealed that *E. coli* used the RHOD PspE to convert thiosulfate to sulfane sulfur, which increased cellular sulfane sulfur from about 100 µM to 200 to 300 µM. A further increase did not occur, possibly due to the product inhibition by GSSH. PspE is a periplasmic protein, and it converts thiosulfate to GSSH in the periplasmic space (Figure 6). GSSH is transported into the cytoplasm via an unknown mechanism. Inside the cytoplasm, it is likely converted to other forms of reactive sulfane sulfur species, including GSSSH, H_2_S_2_, and Cys-SSH; however, GSSH is still the dominant species (Figure 4). Rapid reduction of the increased cellular sulfane sulfur to H_2_S by cellular thiols was not observed, as tested (Appendix A); however, the results cannot rule out the conversion of thiosulfate to sulfane sulfur and then H_2_S in other organisms if they have elevated cellular thiols or sulfane-sulfur-reducing proteins, such as thioredoxin and glutaredoxin. These findings may guide the use of thiosulfate as the donor of H_2_S and sulfane sulfur in studies on the effects of increased cellular sulfane sulfur in bacteria, plants, and animals and in clinical applications.

## Figures and Tables

**Figure 1 antioxidants-12-01127-f001:**
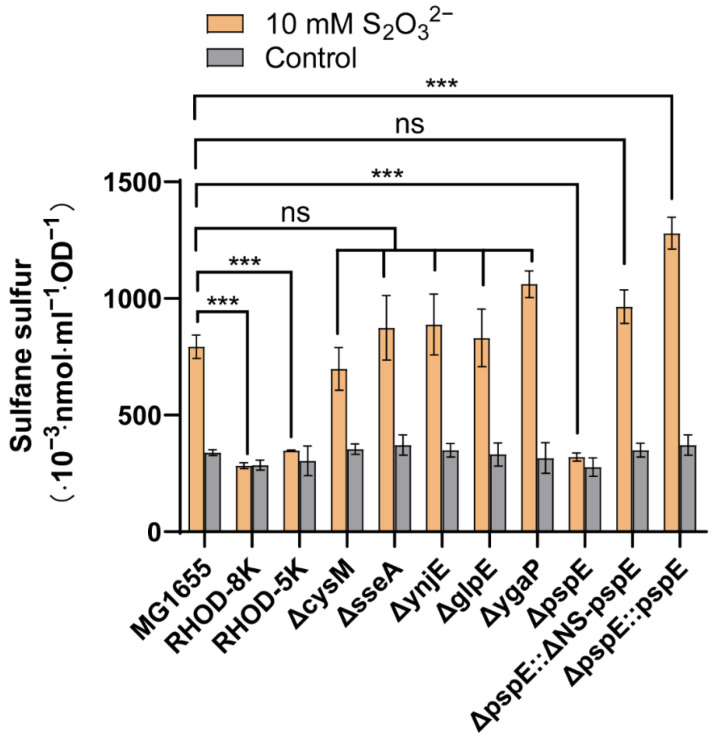
Cellular sulfane sulfur of the *E. coli* MG1655 wild type and RHOD deletion mutants with or without thiosulfate. The *E. coli* MG1655 wild type, mutants, and *pspE*-overexpressed strain were cultured until an OD_600nm_ of 1.0 and then incubated with or without 10 mM thiosulfate for 1 h. The cellular sulfane sulfur reacted with sulfite to produce thiosulfate, which was detected. Three parallel experiments were performed to obtain the averages and standard deviations (*n* = 3). The T-test method was performed to calculate the *p*-values (ns, *p* ≥ 0.05; ***, *p* < 0.001).

**Figure 2 antioxidants-12-01127-f002:**
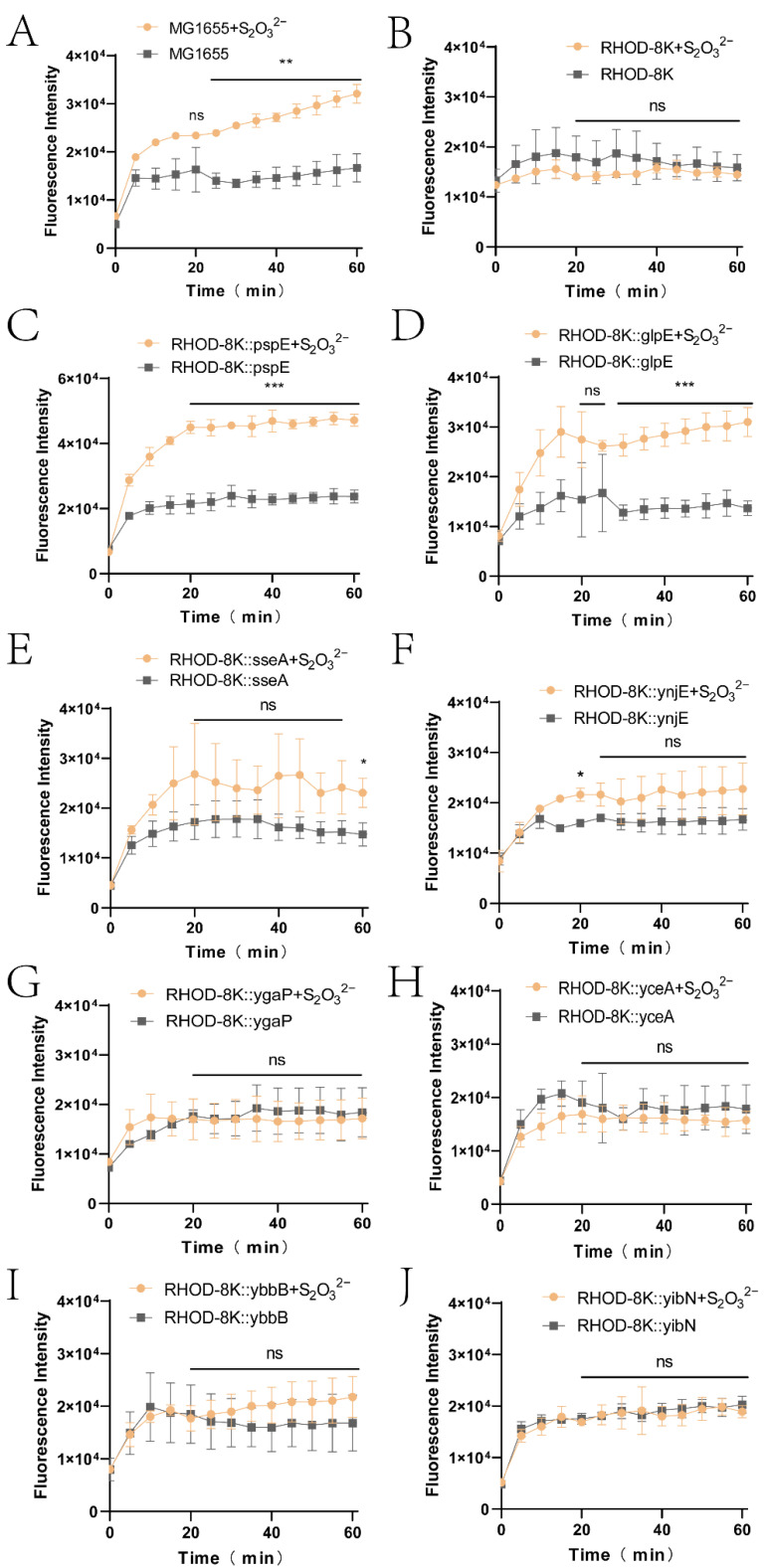
Sulfane sulfur detection using SSP4 during incubation of *E. coli* with thiosulfate. Eight genes encoding RHODs were cloned from the MG1655 wild-type genome, which was ligated with the pBBR1MCS2 plasmid using the T5 exonuclease-dependent assembly method. Plasmids were Ittransformed into RHOD-8K competent cells to produce (**C**) RHOD-8K::pspE, (**D**) RHOD-8K::glpE, (**E**) RHOD-8K::sseA, (**F**) RHOD-8K::ynjE, (**G**) RHOD-8K::ygaP, (**H**) RHOD-8K::yceA, (**I**) RHOD-8K::ybbB, and (**J**) RHOD-8K::yibN with (**A**) MG1655 and (**B**) RHOD-8K as control. The sulfane sulfur content in cells were monitered by using SSP4 staining method in these strains. In brief, the strains were cultured until an OD_600nm_ of 1.0. The cells were harvested and resuspended in PBS buffer at an OD_600nm_ of 0.5. Then, 10 mM thiosulfate was added to the cell, and the control did not receive thiosulfate. After adding 0.5 mM CTAB and 20 μM SSP4, fluorescence was detected using a microplate reader at 37 °C every 5 min for 1 h. CTAB was added to permeabilize the cell membrane. Three parallel experiments were performed to obtain the averages and standard deviations (*n* = 3). The T-test method was performed to calculate the *p*-values (ns, *p* ≥ 0.05; *, *p* < 0.05; **, *p* < 0.01; ***, *p* < 0.001). The subfigures (**A–J**) are used to represent the sulfane sulfur detected in MG1655, RHOD-8K, RHOD-8K::pspE, RHOD-8K::glpE, RHOD-8K::sseA, RHOD-8K::ynjE, RHOD-8K::ygaP, RHOD-8K::yceA, RHOD-8K::ybbB, and RHOD-8K::yibN, respectively.

**Figure 3 antioxidants-12-01127-f003:**
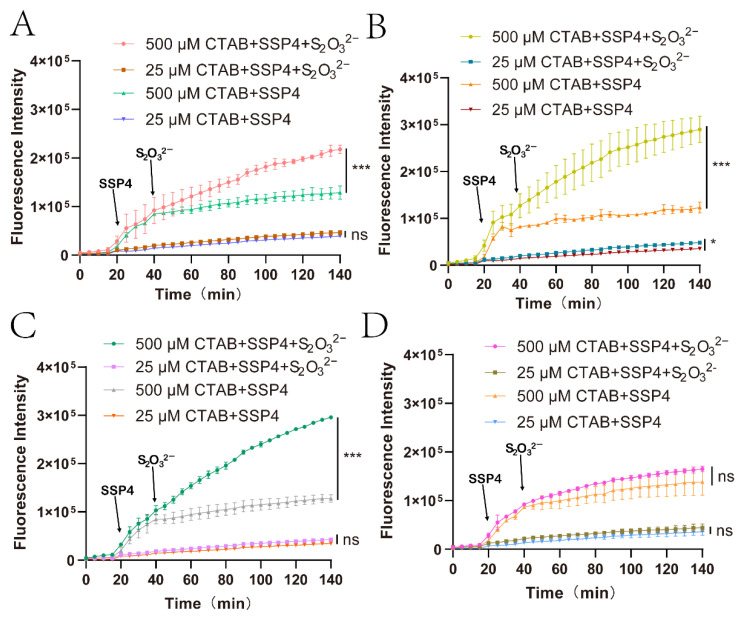
Intracellular and extracellular sulfane sulfur was continuously monitored using SSP4. (**A**) MG1655, (**B**) ΔpspE::pspE, (**C**) ΔpspE::ΔNS-pspE, and (**D**) ΔpspE cells were harvested and resuspended in PBS at an OD_600nm_ of 2.0. An amount of 200 μL of resting cells was transferred to a black 96-well plate with 500 μM or 25 μM CTAB and incubated at 37 °C with shaking. Then, 20 μM SSP4 was added after 20 min, and 10 mM thiosulfate was added after 40 min. The excitation wavelength (Ex) and emission wavelength (Em) at 482 nm and 515 nm were continuously detected every 5 min for 2.3 h. Three parallel experiments were performed to obtain the averages and standard deviations (*n* = 3). The T-test method was performed to calculate the *p*-values (ns, *p* ≥ 0.05; *, *p* < 0.05; ***, *p* < 0.001).

**Figure 4 antioxidants-12-01127-f004:**
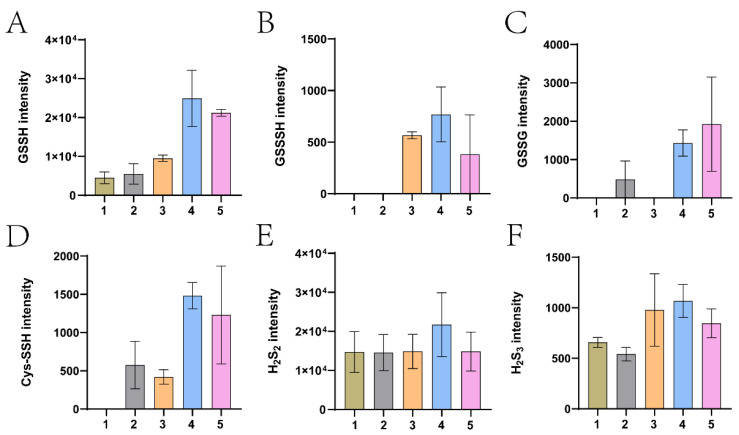
LC-MS analysis of cellular sulfane sulfur species in *E. coli*. The *E. coli* strains were cultured until an OD_600nm_ of 1.0 before being harvested and resuspended in PBS. Resting cells at an OD_600nm_ of 2.0 were incubated with 10 mM thiosulfate for 1 h, and then harvested. Cellular sulfane sulfur species were analyzed using LC-MS. (**A**) GSSH, (**B**) GSSSH, (**C**) GSSG, (**D**) Cys-SSH, (**E**) H_2_S_2_, and (**F**) H_2_S_3_ were detected. 1, the control MG1655 without thiosulfate; 2, ΔpspE + thiosulfate; 3, MG1655 + thiosulfate; 4, ΔpspE::pspE + thiosulfate; 5, ΔpspE::ΔNS-pspE + thiosulfate. Two parallel experiments were performed, with averages and ranges being presented.

**Figure 5 antioxidants-12-01127-f005:**
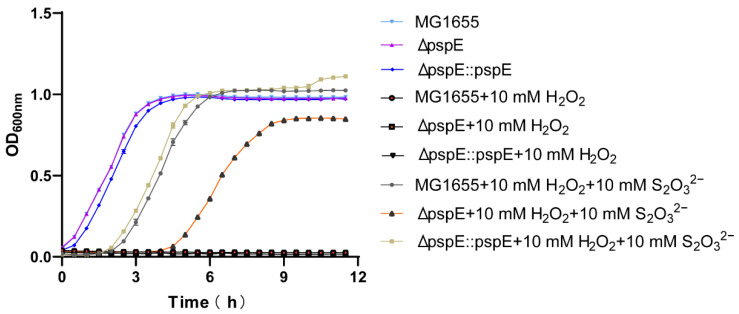
Growth curves of *E. coli* when incubated with thiosulfate and hydrogen peroxide. The MG1655 wild type, ΔpspE, and ΔpspE::pspE were incubated in 400 μL LB medium at an initial OD_600nm_ of 0.05 in 48-well plates. No growth was observed for all strains with 10 mM H_2_O_2_. With 10 mM H_2_O_2_ and 10 mM thiosulfate, delayed growth was observed. Three parallel experiments were performed to obtain the averages and standard deviations (*n* = 3).

**Figure 6 antioxidants-12-01127-f006:**
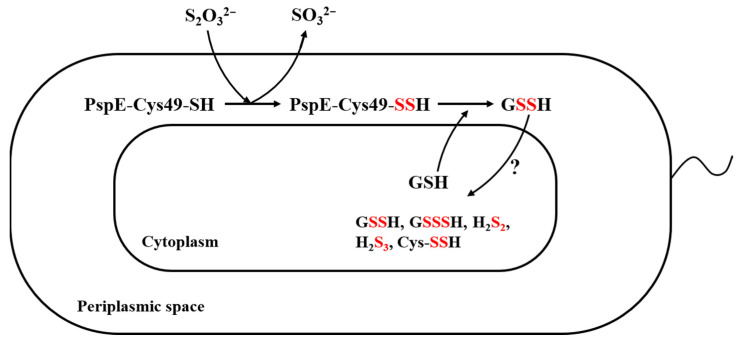
The proposed conversion of thiosulfate to cellular sulfane sulfur in *E. coli*. PspE converts thiosulfate to GSSH in the periplasmic space, which is then transported into the cytoplasm via an unknown mechanism. The detected cellular sulfane sulfur species include GSSH, GSSSH, H_2_S_2_, H_2_S_3_, and Cys-SSH.

**Table 1 antioxidants-12-01127-t001:** Kinetic parameters of several *E. coli* RHODs.

RHODs	*K_m_* (GSH) mM	*K_m_* (S_2_O_3_^2−^) mM	Vmax nmol·min^−1^·mg^−1^	k_cat_ s^−1^	k_cat_/*K_m_* (GSH) M^−1^ s^−1^	k_cat_/*K_m_* (S_2_O_3_^2−^) M^−1^ s^−1^
ΔNS-PspE	28.9 ± 3.0	7.8 ± 1.6	212.7 ± 13.9	0.04	1.3	4.8
GlpE	14.3 ± 0.1	33.6 ± 7.0	109.1 ± 12.2	0.02	1.7	0.7
ΔNS-YnjE	-	-	-	-	-	-
YceA	-	-	-	-	-	-

Note: The kinetic parameters were assayed with either fixed GSH or thiosulfate at 100 mM and varying concentrations of the other substrate. Three parallel experiments were performed to obtain the averages and standard deviations (*n* = 3). The symbol “-” indicates that no activity was detectable. ΔNS-PspE, GlpE, ΔNS-YnjE, and YceA in the reaction mixtures were used at 5, 5, 200, and 100 μg/mL, respectively.

**Table 2 antioxidants-12-01127-t002:** The persulfidation of ΔNS-PspE Cys49 after incubating with thiosulfate.

Sequence	Protein	Modification	Intensity	Percentage (%)
IATAVPDKNDTVKVYCNAGR	ΔNS-PspE + S_2_O_3_^2−^	Cys-SH	7,288,433,560	98.9
		Cys-SSH	79,902,321	1.1
	ΔNS-PspE	Cys-SH	121,956,774	100
		Cys-SSH	0	0

## Data Availability

All data are reported in the main text or Appendix A.

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
