# Peer review of "The Rhodanese PspE Converts Thiosulfate to Cellular Sulfane Sulfur in Escherichia coli"

_antioxidants, 2023, doi:10.3390/antiox12051127_

Round 1
Reviewer 1 Report
Review of the paper entitled „ The rhodanese PspE converts thiosulfate to cellular sulfane sulfur in Escherichia coli by Qiaoli Yu, Mingxue Ran, Yuping Xin, Huaiwei Liu, Honglei Liu, Yongzhen Xia and Luying Xun.
The authors have previously reported that thiosulfate is an effective sulfane sulfur donor in Escherichia coli (E. coli). In this study they showed that one of the multiple rhodaneses, PspE, is responsible for the conversion of thiosulfate to sulfane sulfur in E.coli cells.
This paper is interesting.
I have some comments.
Introduction
The introduction is well written but needs some improvement.
The abbreviation PDO should be defined.
The cysteinyl-tRNA synthetase catalyze co-translational cysteine polysulfidation, so it is not (in my opinion) an enzyme involved in cysteine metabolism unlike 3-mercaptopuruvate sulfurtransferase, thiosulfate sulfurtransferase (rhodanese) and cystathionine γ-lyase.
„RHODs are referred as thiosulfate:cyanide sulfurtransferases (E.C.2.8.1.1), as they detoxify cyanide”. Indeed, the systematic name of this enzyme class is thiosulfate:cyanide sulfurtransferase but the two substrates of this enzyme are also thiosulfate and glutathione.
Generally, rhodanase transfers sulfur atoms from anionic donors (compounds containing sulfane sulfur) to various nucleophilic acceptors. The authors should characterize this class of enzymes in more detail here. Please take note of my comment regarding the Material and Methods section.
„Thus, thiosulfae has a good potential to be used as a donor of H2S and sulfane sulfur”. Where? In plants, in mammals, in animals, in bacteria.... ?
Materials and Methods
One of the titles is Thiosulfate:GSH Sulfurtransferase Activity Assay. The authors characterize rhodanase in the Introduction as thiosulfate:cyanide sulfurtransferase. It should be introduced in the Introduction that rhodanase is defined differently, depending on the substrates, and it should be noted that if the substrates are thiosulfate and glutathione (GSH), then this enzyme is referred as Thiosulfate:GSH Sulfurtransferase.
Why did the authors use 0.5 mM - 150 mM thiosulfate and 0.5 mM - 150 mM GSH? Were these studies aimed at measuring enzyme activity or determining its kinetic parameters. It's not clear.
The authors wrote that „lead sulfite was measured at 390 nm”. The lead sulfide is a sparingly soluble salt which forms a black precipitate. How can it be measured?
In the subsection entitled “HPLC Analysis of Total Cellular Sulfane Sulfur” the authors wrote “Thiosulfate was added into cell culture at room temperature for 1 h. Total cellular sulfane sulfur was converted to stable thiosulfate before being analyzed, according to a reported method. I think there is some error here. Because if the authors assumed that total cellular sulfane sulfur was converted to stable thiosulfate, then for what purpose did they add thiosulfate to the cells earlier. If so, the authors measured the sum of thiosulfate produced in the cells and added by the authors, but why was this thiosulfate added? This should be clarified.
The subsection entitled “SSP4 Staining for Sulfane Sulfur Detection”.
How long was the incubation of “E. coli resting cells in PBS at OD600nm of 2.0 with 20 μM SSP4 at 37oC in the dark”.
The subsection entitled “GSSH Preparation and detection”
GSSG was determined by the authors using a cyanide method. In my opinion, this is not a selective method. In addition to GSSH, chemical compounds such as GSSSG or S8 can also react. Why the authors did not use mass spectrometry here?
The subsections entitled “HPLC Analysis for sulfide detection” and “LC-MS/MS Analysis PspE-Cys49”.
The authors use thiosulfate at a concentration of 10 mM. On what basis the authors chose this concentration? In my opinion different concentrations of thiosulfate should be tested.
Results
Figure 1. In the caption under this Figure, the authors wrote „Cellular sulfane sulfur of E. coli ....... with or without thiosulfate” (my underlining). In the caption under this Figure, the authors also wrote „The cellular sulfane sulfur was reacted with sulfite to produce thiosulfate (my underlining ) which was detected”. Something doesn't add up here.
Figure 2. The authors should explain why they added CTAB.
Table 3. Are you sure the concentration of glutathion was 100 mM? Please also consider my earlier comment regarding Material and Methods, second paragraph.
Figure 3. Are the authors sure that the reagents alone (SSP4 + thiosulfate) without cells do not have any fluorescence? It might be worth considering making a so-called reagent blank.
Discussion
„Since most E. coli RHODs failed to produce cellular sulfane sulfur from thiosulfate, they may not catalyze the reaction of thiosulfate:GSH sulfurtransferase”. Where does this view come from? Is it based on previous research by the authors? Or possibly on literature data? Authors should cite the relevant publication here. Does the sulfur atom transfer reaction from thiosulfate to cyanide occur in E. coli cells?
The part of the Discussion that begins with the paragraph "H2S has numerous physiological effects. It is...” (line 427) should be shortened, because it does not concern the results obtained by the authors, only the biological properties of H2S in the cells of various living organisms.
General
The authors should be consistent in writing abbreviations, names, etc
For example. Line 45 –PDO, line 417 – persulfide dioxygenase. Line 361 – SseA, Line 361 – sseA. Line 367 – pspE, line 381 (and other) – PspE. Line 43 (and other) – L-cysteine, line 49 (and other) L-cysteine. On line 465, instead of „the rhodanese PspE”, it should be the RHOD PspE. In my opinion just write Cys instead of Cys-thiol (line 441 and other).
The entire manuscript requires thorough language correction due to numerous grammatical and syntax errors, typos, etc. For example. Line 182. Do not use the imperative. Moreover; is „cluture”; should be „culture”. „Our data suggest that E. coli mainly uses PspE to convert thiosulfate to cellular sulfane sulfur is partly because it is a major RHOD in E. coli” (line 383). In my opinion it should be Our data suggest that E. coli 382 mainly uses PspE to convert thiosulfate to cellular sulfane sulfur is partly because it is a major RHOD in E. coli .
So, these are just some examples, there are more errors. The manuscript requires a thorough language proofreading .

The authors should be consistent in writing abbreviations, names, etc
For example. Line 45 –PDO, line 417 – persulfide dioxygenase. Line 361 – SseA, Line 361 – sseA. Line 367 – pspE, line 381 (and other) – PspE. Line 43 (and other) – L-cysteine, line 49 (and other) L-cysteine. On line 465, instead of „the rhodanese PspE”, it should be the RHOD PspE. In my opinion just write Cys instead of Cys-thiol (line 441 and other).
The entire manuscript requires thorough language correction due to numerous grammatical and syntax errors, typos, etc. For example. Line 182. Do not use the imperative. Moreover; is „cluture”; should be „culture”. „Our data suggest that E. coli mainly uses PspE to convert thiosulfate to cellular sulfane sulfur is partly because it is a major RHOD in E. coli” (line 383). In my opinion it should be Our data suggest that E. coli 382 mainly uses PspE to convert thiosulfate to cellular sulfane sulfur is partly because it is a major RHOD in E. coli .
So, these are just some examples, there are more errors. The manuscript requires a thorough language proofreading .
Author Response
Reviewer I
Introduction
POD is defined.
Reworded by removing “metabolism”. “Besides from H2S oxidation, sulfane sulfur can be produced from cysteine (metabolism) by various enzymes, including cysteinyl-tRNA synthetase, 3-mercaptopyruvate sulfurtransferase, cystathionine γ-lyase [9-12].”
RHOD: RHODs are originally referred as thiosulfate:cyanide sulfurtransferases (E.C.2.8.1.1), as they detoxify cyanide, and now they are known to transfer sulfur atoms from compounds containing sulfane sulfur to various nucleophilic acceptors [38,39].
Thus, thiosulfae has a good potential to be used as a donor of H2S and sulfane sulfur in mammals and bacteria.
Materials and Methods
The RHOD activity as thiosulfate:GSH sulfurtransferase as defined.
“When RHOD uses thiosulfate and GSH as the substrates, it is referred as Thiosulfate:GSH Sulfurtransferase, producing GSSH and sulfite; GSSH spontaneously reacts with another GSH to generate GSSG and H2S. H2S was detected by including lead-acetate in the reaction mixture, as previously described [50].”
Why did the authors use 0.5 mM - 150 mM thiosulfate and 0.5 mM - 150 mM GSH?
A: The Km values (about 30 mM) are relatively high, we tried to use the concentrations 5 times of the Km as normally the kinetic parameters are determined.
The Authors wrote that „lead sulfite was measured at 390 nm”. The lead sulfide is a sparingly soluble salt which forms a black precipitate. How can it be measured?
A: Yes. The precipitates are fine particles, possibly nanoparticles, as clear particles are not visible with eyes. The method has been used by Dr. xx’s group [50].
In the subsection entitled “HPLC Analysis of Total Cellular Sulfane Sulfur” the authors wrote “Thiosulfate was added into cell culture at room temperature for 1 h. Total cellular sulfane sulfur was converted to stable thiosulfate before being analyzed, according to a reported method. I think there is some error here. Because if the authors assumed that total cellular sulfane sulfur was converted to stable thiosulfate, then for what purpose did they add thiosulfate to the cells earlier. If so, the authors measured the sum of thiosulfate produced in the cells and added by the authors, but why was this thiosulfate added? This should be clarified.
A: Here, the cells were washed, and cellular sulfane sulfur reacted with sulfite to produce thiosulfate, which was derivatized and analyzed by using HPLC. A brief description was added with citation. “Thiosulfate was added into cell culture at room temperature for 1 h. Total cellular sulfane sulfur was determined according to a reported method [46]. Briefly, the cells were harvested by centrifugation, washed once with 50 mM Tris buffer (pH 8.0), and resuspended in 50 mM Tris buffer (pH 9.5) with 1 mM sulfite. The sample was heated in boiling water for 10 min, during which cellular sulfane sulfur reacted with sulfte to produce thiosulfate. The produced thiosulfate was derivatized with mBBr and then analyzed by using HPLC.”
The subsection entitled “SSP4 Staining for Sulfane Sulfur Detection”.
How long was the incubation of “E. coli resting cells in PBS at OD600nm of 2.0 with 20 μM SSP4 at 37oC in the dark”.
A: For 1 or 2 h with measuring at 5-min intervals (Fig. 2&3).
GSSH Preparation and detection
Where did you used GSSH in the manuscript?
The authors use thiosulfate at a concentration of 10 mM. On what basis the authors chose this concentration? In my opinion different concentrations of thiosulfate should be tested.
A: Yes. Different concentrations have been used and reported [xx]. In the previous paper, 10 mM was shown to be not toxic to E. coli, but enhanced cellular sulfane sulfur. Here, we tried to figure out how thiosulfate is converted to cellular sulfane sulfur.
Results
Figure 1. In the caption under this Figure, the authors wrote „Cellular sulfane sulfur of E. coli ....... with or without thiosulfate” (my underlining). In the caption under this Figure, the authors also wrote „The cellular sulfane sulfur was reacted with sulfite to produce thiosulfate (my underlining ) which was detected”. Something doesn't add up here.
A: As explained in the method section, cells were harvested and washed to remove thiosulfate in the medium. Cellular sulfane sulfur was then reacted with sulfite to produce sulfite to produce thiosulfate, which was derivatized and analyzed. The method is effective to determine cellular sulfane sulfur and has been reported [Ran; Yu]
Figure 2. The authors should explain why they added CTAB.
A: CTAB was added to permeabilize the cell membrane.
Table 3. Are you sure the concentration of glutathion was 100 mM? Please also consider my earlier comment regarding Material and Methods, second paragraph.
A: Yes. The Km values were around 30 mM, and 5 times of the Km value was used to determine the kinetics.
Figure 3. Are the authors sure that the reagents alone (SSP4 + thiosulfate) without cells do not have any fluorescence? It might be worth considering making a so-called reagent blank.
A: Yes. We have done the experiment, and SSP4 does not react with thiosulfate.
„Since most E. coli RHODs failed to produce cellular sulfane sulfur from thiosulfate, they may not catalyze the reaction of thiosulfate:GSH sulfurtransferase”. Where does this view come from? Is it based on previous research by the authors? Or possibly on literature data? Authors should cite the relevant publication here. Does the sulfur atom transfer reaction from thiosulfate to cyanide occur in E. coli cells?
A: The results are reported in Fig. 2 and Table 3. The sentence is revised as follows: “Since most E. coli RHODs failed to complement the E. coli RHOD-8K mutant to produce cellular sulfane sulfur from thiosulfate (Fig. 2),”
General
Authors should be consistent in writing abbreviations, names, etc
For example. Line 45 –PDO, line 417 – persulfide dioxygenase. Line 361 – SseA, Line 361 – sseA. Line 367 – pspE, line 381 (and other) – PspE. Line 43 (and other) – L-cysteine, line 49 (and other) L-cysteine. On line 465, instead of „the rhodanese PspE”, it should be the RHOD PspE. In my opinion just write Cys instead of Cys-thiol (line 441 and other).
A: Full name of PDO is used and defined when it is first mentioned (Line x). SseA is used to indicate the protein, and sseA indicates the gene. The approaches are common in journals of microbiology.
The entire manuscript requires thorough language correction due to numerous grammatical and syntax errors, typos, etc. For example. Line 182. Do not use the imperative. Moreover; is „cluture”; should be „culture”. „
A: Changed to “One mL of the culture was centrifuged at 13,000 × g for 3 min at 4oC.”
Our data suggest that E. coli mainly uses PspE to convert thiosulfate to cellular sulfane sulfur is partly because it is a major RHOD in E. coli” (line 383). In my opinion it should be Our data suggest that E. coli 382 mainly uses PspE to convert thiosulfate to cellular sulfane sulfur is partly because it is a major RHOD in E. coli .
A: Revised to “Thus, E. coli primarily uses PspE to convert thiosulfate to cellular sulfane sulfur.”
So, these are just some examples, there are more errors. The manuscript requires a thorough language proofreading.
A: We carefully read the manuscript sentence by sentence and made additional corrections.
Reviewer 2 Report
Manuscript entitled “The rhodanese PspE converts thiosulfate to cellular sulfane sulfur in Escherichia coli”, submitted by Qiaoli Yu, Mingxue Ran, Yuping Xin, Huaiwei Liu, Honglei Liu, Yongzhen Xia, Luying Xun, can be considered for publication in Antioxidants Journal, after a major revision.
Here is a list of my specific comments:
1. Page 2, line 55: “Various H2S donors have been developed…”. Add here more references.
2. Page 2, line 82: “Here, we show that the RHOD…”. At the end of Introduction, the main objectives of this study should be clearly and detailed presented.
3. Page 5, line 123: “…indicated in the text.” The variation interval should be added here.
4. Page 13, 4. Discussion: In this section provide a detailed discussion of the experimental results presented in the previous section. Delete irrelevant observations/comments. Pay attention on the interpretation of all experimental results.
5. Page 15, 5. Conclusion: This section should be detailed. Include in this section the most important experimental results and findings to highlight the importance of this study.
6. Page 15, Figure 6: This figure is not mentioned in the text.
7. Page 16, References: The number of references should be reduced.
Author Response
Here is a list of my specific comments:
- Page 2, line 55: “Various H2S donors have been developed…”. Add here more references.
A: We added 2 review articles as references.
- Page 2, line 82: “Here, we show that the RHOD…”. At the end of Introduction, the main objectives of this study should be clearly and detailed presented.
A: Revised to “Here, we used genetic and biochemical analysis and showed that the RHOD PspE is mainly responsible for rapidly converting stable thiosulfate to reactive sulfane sulfur in E. coli.”
- Page 5, line 123: “…indicated in the text.” The variation interval should be added here.
A: every 10 min.
- Page 13, 4. Discussion: In this section provide a detailed discussion of the experimental results presented in the previous section. Delete irrelevant observations/comments. Pay attention on the interpretation of all experimental results.
A: Discussion is revised and reduced.
- Page 15, 5. Conclusion: This section should be detailed. Include in this section the most important experimental results and findings to highlight the importance of this study.
A: Conclusion was revised according to the suggestion.
- Page 15, Figure 6: This figure is not mentioned in the text.
A: Figure 6 is cited in the revised conclusion.
- Page 16, References: The number of references should be reduced.
A: Reduced.
Reviewer 3 Report
This article is covering some aspects of interaction of rhodanese PspE to converts thiosulfate to cellular sulfane sulfur in Escherichia coli.
The specific aims of this article are exclusively directed to investigate the level of H2S and elevated cellular sulfane sulfur and the speculation on elevated cellular thiols or sulfane sulfur reducing proteins, such as thioredoxin and glutaredoxin.
This will constitute the important goals and novelty of this paper. The article is concluded with a collection of 85 mostly recent references. Additionally, all 6 Figures and 4 Tables are informative and with concise important data comparison along with the kinetic parameters.
The following suggested changes and recommendations should be introduced before the publication of the manuscript.
1. Page 1. Line 38. Insert “anion” after Thiosulfate
2. Page 2. Line 55. Correct misspelling “theropeutic” to “therapeutic”
3. Page 2. Line 78. Correct misspelling “sulfuane” to “sulfane”
4. Page 4. Line 102. Remove “briefly” as it is not in correct grammatical format.
5. Page 7. Line 219. Move here figure 1 in order to expose the important information.
6. Page 13. Line 359. Insert “way” after “In the first”.
7. Page 14. Line 393. Insert “consequently after (Figure 3).
8. Page 15. Line 443. Replace ”oxidation production” with “ formation. Insert “and” after H2S.
9. Page 16. Line 447. Double abbreviation “GSSH” remove the second “GSSH”
10. It is critically important to insert diagrams on sulfane sulfur and chemical structure of GSSH in the 1. Introduction as depicted below:
11. References: Important reference on PspE of Escherichia coli is missing and must be added to the references as:
PspE (phage-shock protein E) of Escherichia coli is a rhodanese.
Hendrik Adams, Wieke Teerstra, Margot Koster, Jan Tommassen, FEBS Letters 518 (2002) 173-176.
The manuscript is of average quality and importance and is written and edited in order to meet the standard for the articles published inAntioxidants. Thus, I certainly recommend it for publication after the correction of these suggested minor changes and recommendations.
English language will require professional technical correction!!!
Author Response
This article is covering some aspects of interaction of rhodanese PspE to converts thiosulfate to cellular sulfane sulfur in Escherichia coli.
The specific aims of this article are exclusively directed to investigate the level of H2S and elevated cellular sulfane sulfur and the speculation on elevated cellular thiols or sulfane sulfur reducing proteins, such as thioredoxin and glutaredoxin.
This will constitute the important goals and novelty of this paper. The article is concluded with a collection of 85 mostly recent references. Additionally, all 6 Figures and 4 Tables are informative and with concise important data comparison along with the kinetic parameters.
The following suggested changes and recommendations should be introduced before the publication of the manuscript.
- Page 1. Line 38. Insert “anion” after Thiosulfate
- Page 2. Line 55. Correct misspelling “theropeutic” to “therapeutic”
- Page 2. Line 78. Correct misspelling “sulfuane” to “sulfane”
- Page 4. Line 102. Remove “briefly” as it is not in correct grammatical format.
- Page 7. Line 219. Move here figure 1 in order to expose the important information.
- Page 13. Line 359. Insert “way” after “In the first”.
- Page 14. Line 393. Insert “consequently after (Figure 3).
- Page 15. Line 443. Replace ”oxidation production” with “ formation. Insert “and” after H2S.
- Page 16. Line 447. Double abbreviation “GSSH” remove the second “GSSH”
Thank you for the suggestions. Revised.
Reviewer 4 Report
Introduction. The objectives of the study must be described clearly.
Materials and methods.
Please provide details of isolation circumstances for all the strains (as supplementary material)
All the primers must be shown as supplementary material, not in the main text.
For all the PCRs, it is paramount to also show all the details (temperature, product size etc.), not just the primers.
Results
All the figures must be colorized.
Figure 6 is not mentioned in the text.
Discussion.
1. This is very long and thus must be divided in two or three parts to allow better flow of the text.
2. Please add a new sub-section with the clinical implications of the findings.
3. Some recent (February to April 2023) references are missing.
Extensive editing of English language required.
Author Response
Introduction. The objectives of the study must be described clearly.
A: Added in the last paragraph of the introduction. “Here, we used genetic and biochemical analysis and showed that the RHOD PspE is mainly responsible for rapidly converting stable thiosulfate to reactive sulfane sulfur in E. coli.”
Materials and methods.
Please provide details of isolation circumstances for all the strains (as supplementary material)
A: Move Table 1 to supplemental (Table S1). The new strains and mutants were given in the method section.
All the primers must be shown as supplementary material, not in the main text.
A: Move Table 2 to supplemental (Table S2).
For all the PCRs, it is paramount to also show all the details (temperature, product size etc.), not just the primers.
A: The information is given either in the new Table S2 or as a footnote.
Results
All the figures must be colorized.
A: Colored figures were used in the revised manuscript. A new graphic abstract was prepared.
Figure 6 is not mentioned in the text.
A: Cited in the conclusion.
Discussion.
- This is very long and thus must be divided in two or three parts to allow better flow of the text.
A: The discussion section is reduced and revised.
- Please add a new sub-section with the clinical implications of the findings.
A: The use of thiosulfate as a reactive sulfur donor in clinical implications are discussed in the end of the discussion and the conclusion.
- Some recent (February to April 2023) references are missing.
A: We added two references published in 2023.
Extensive editing of English language required.
A: The manuscript has been carefully read and revised. The final version was checked by Grammarly.
Round 2
Reviewer 1 Report
The authors have greatly improved their manuscript. In my opinion, this version of this paper can be accepted for publication.
Author Response
Thank you!
Reviewer 2 Report
All my previous remarks and comments have been considered in this new version of the manuscript. In my opinion, the revised manuscript meets the criteria and can be published as original paper in Antioxidants Journal.
Author Response
Thank you!
Reviewer 4 Report
Before final acceptance, the authors must add a brief passage about the clinical implications of their study.
Moderate editing of English language.
Author Response
Before final acceptance, the authors must add a brief passage about the clinical implications of their study.
A: We talked about H2S and sulfane sulfur donors in clinical application. In the same paragraph (last one in the discussion), we added: “Our finding that the RHOD PspE converts thiosulfate to increase cellular sulfane sulfur in E. coli may guide the use of thiosulfate as a donor of reactive sulfane sulfur in clinical applications.”
Further discussion on clinical applications is beyond the scope of this manuscript and our capability, as we have not experience on the topic.